# Geometric Effect on the Nonlinear Force-Displacement Relationship of Awl-Shaped Serpentine Microsprings for In-Plane Deformation

**DOI:** 10.3390/ma13122864

**Published:** 2020-06-26

**Authors:** Meng-Ju Lin, Hui-Min Chou, Rongshun Chen

**Affiliations:** 1Department of Mechanical and Computer-Aided Engineering, Feng Chia University, Taichung City 407, Taiwan; 2Department of Power Mechanical Engineering, National Tsing Hua University, Hsinchu City 300, Taiwan; d9933827@oz.nthu.edu.tw (H.-M.C.); rchen@pme.nthu.edu.tw (R.C.)

**Keywords:** awl-shaped, serpentine spring, MEMS, hardening spring

## Abstract

Even when made by brittle materials, awl-shaped serpentine microsprings (ASSMs) were found to have a nonlinear displacement–force relationship similar to springs made by ductile material. It is found that the nonlinear displacement–force relationship is due to the geometry and dimensions of the ASSMs. The geometric effect of the nonlinear force–displacement relationship of ASSMs for in-plane motion was investigated. A theoretical solution was derived to analyze this nonlinearity. By successfully fabricating and measuring an ASSM, the theoretical results agreed well with the experimental results. The results indicated that ASSMs have a nonlinear force–displacement relationship, which is similar to that of hardening springs. The taper angle has a significant effect on the nonlinear displacement of ASSMs. When the taper angle was small, no obvious effect appeared on the nonlinearity of the microsprings with different numbers of turns. When the beam length increased, the critical force for nonlinear deflection decreased.

## 1. Introduction

Microelectromechanical systems (MEMSs) are fully developed and allow for tremendous applications. In MEMS devices, microsprings are crucial for the mechanical structure because they reduce vibration, offer precise positions, are connected to other components, apply elastic force, and support the structure of the device. Therefore, the microsprings in MEMS devices have many applications. Microsprings can be used in MEMS actuators [1,2,3,4,5], MEMS sensors [6,7,8], in the micro-2D stage (micro-X–Y stage) [9,10,11,12], and for energy harvesting [13,14]. Improving the performance of microsprings is a key research topic. Performance includes supporting stability [15], improving reliability [16], cross-axis coupling [17], and staging in-plane and out-of-plane motion [18]. Therefore, different types of microsprings [19,20,21,22] have been designed. The types of microsprings proposed in the literature are folded beam-type springs [21], double-clamped beam-type springs [23], crab-leg springs [24], serpentine springs [25,26], W-form springs [6], L-shape planar springs [27], and conical springs [28]. Among these types of microsprings, serpentine springs could be used in out-of-plane and in-plane motion. Serpentine springs can provide larger deflection for their lower stiffness within an appropriate and reasonable area of MEMS device layouts. Therefore, they are widely used in MEMS devices. For microserpentine investigation, Barillaro et al. designed two types of serpentine springs and analyzed their spring constants for out-of-plane motion through theoretical analysis and numerical analysis by using the finite element method [22]. Liu et al. discussed the stress and stiffness of S-shaped springs and reported that the number of turns of springs has a significant effect on their stress and stiffness [29]. Yeh et al. found that serpentine springs can provide a larger rotating angle than the springs of clamped straight beams in comb drive actuators [30]. Su et al. discussed the design principles for highly reliable serpentine springs for MEMS optical switches involving large mirror mass [31]. Sharma and Gupta [32] designed a nonuniform serpentine spring for radio frequency MEMS switches; this design reduced actuating voltage. Chou, Lin, and Chen designed and analyzed the mechanical properties of awl-shaped serpentine microsprings (ASSMs) for in-plane and out-of-plane motion [33,34]. They found that under the same total effective lengths and folds, for the out-of-plane motion, the ASSMs had a smaller spring constant to layout area ratio than the traditional serpentine microsprings. For in-plane motion, the parameter spring constant to layout area ratio (*k/A*) is used as the performance index for ASSMs. The ASSMs had a smaller spring constant to layout area ratio than the traditional serpentine microsprings. Furthermore, they reported that the spring constant of the in-plane motion was always larger than that of the out-of-plane motion when the width was larger than the thickness. If the width was smaller than the thickness, and the taper angle had a value that was more than critical, the spring constant of the in-plane motion was less than that of the out-of-plane motion.

It is well known that ductile springs have a nonlinear displacement–force relationship because the displacement is sufficiently large. The nonlinear relationship between the force and displacement is often expressed by the following equation:(1)$$F=k_{1}x+k_{3}x^{3}$$
where *F* is applied force, *x* is displacement, and *k_1_* and *k_3_* are constants. If *k*_3_ > 0, the spring is called a “hardening spring”. When *k*_3_ < 0, the spring is called a “softening spring”. Traditionally, springs made from ductile materials such as metal present nonlinear behavior such as the hardening spring. However, in MEMSs, the springs are often produced from brittle materials such as silicon. It is known that the stress–strain curve of brittle materials is almost linear before the yield point, and the brittle material would fracture after the yield point; there is almost no nonlinear region. Therefore, whether they have a nonlinear force–displacement relationship like a hardening spring will be investigated in this work. Due to the linear stress–strain relationship of the brittle materials, their deflection material properties would be linear. Therefore, it could be assumed that the nonlinear force–displacement relationship of ASSMs may be caused by the structure dimensions and geometry. Chou, Lin, and Chen discussed the geometric effect of ASSMs on nonlinearity for in-plane motion. They used numerical analysis through a finite element method simulation to discuss this effect [35]. They found that the taper angle had a significant effect and the number of turns of the spring had no obvious effect. However, they have no theoretical solution for detailed analysis of nonlinearity for the in-plane motion of ASSMs. A literature review indicated that the nonlinear relationship of ASSMs has seldom been discussed. In this work, the nonlinear relationship in ASSMs was investigated. A theoretical solution to the relationship between the force application and deflection was obtained. The ASSMs were successfully fabricated through silicon-based micromachining, and the theoretical results agreed well with the experimental results. The material of the ASSMs is single crystalline silicon. It has good mechanical properties and is widely used in MEMS structures.

## 2. Materials and Methods

The ASSM model is displayed in Figure 1. The spring comprises a series of vertical and horizontal beams. The vertical beam is a short beam used to connect the horizontal long beam. The horizontal beam is a long beam that is deflected by bending moments. All the short beams have the same length of *p*. The long beams have increasing lengths of *l* + *nt* due to the taper angle *φ*, in which *t = p ×* tan*φ* and *n* is the number of turns of the spring. The beam has a rectangular cross-sectional area. The beam has a width of *w* and thickness of *h*. An ASSM becomes a traditional serpentine microspring when *φ =* 0°. As tensile force is applied on the top vertical short beam and the anchor at the bottom (*y*-direction), the total spring tenses from the bending horizontal long beams.

The free body diagram is presented in Figure 2. In the ASSM free body diagram, the structure is broken into series beams. The series beams connection is a vertical beam followed by a horizontal beam. The free end of the microspring is the short beam that applies force, and the other short beam is constrained to the anchor end. The short vertical beams (*y*-direction) have strong rigidities in the *y*-direction. Therefore, only the *y*-direction deflections of the long beams from the bending moment were considered.

The nonlinear force–displacement relationship of serpentine springs such as hardening springs was considered because of the large deflection of the beam from the bending moment when increased forces were applied. For the ASSM in this work, the short beam length *p* is less than the thickness *h*. Therefore, the deformation by bending moment of the short beam can be neglected. Due to the short beam having strong rigidity and negligible deformation, the deflection of the long beams through the bending moment was considered for the cantilever beams. The large deflection of the cantilever beams from the bending moment was theoretically derived from the fundamental Euler–Bernoulli theorem [36] and integral approach [37].

For concentrated transverse loads, the cantilever beam bends, and the deflection is displayed in Figure 3. The beam without a load applied has an initial length of *l*. After applying the load on the free end, the cantilever beam bends and has a projective length of *l_e_*. Angle *θ* is the rotating angle of the deformed beam at the free end. From the integral approach [37], the angle *θ* can be expressed as follows:(2)$$\sin\theta =\frac{F}{EI}\frac{l_{e}^{2}}{2}$$

The Equation (2) can be rewritten as
(3)$$\frac{l_{e}}{l}=\sqrt{2}sin^{1/2}\theta /{\left(\frac{Fl^{2}}{EI}\right)}^{1/2}$$
where *F* is the applied force, *E* is Young’s modulus, and $I=\frac{1}{12}wh^{3}\;$ is the moment of inertia. Solving Equations (2) and (3) requires a numerical solution. However, compared to the beam length, the large displacement of brittle materials from the bending moment is small enough. Therefore, the rotating angle *θ* of the deformed beam at the free end is also very small. The small *θ* results in projective length *l_e_* being *l_e_* ≈ *l*. When the displacement by bending moment is small compared to the beam length, the rotating angle *θ* of the deformed beam at the free end can be expressed as [38]:(4)$$\theta =\frac{Fl^{2}}{2EI}$$

Taylor’s series of sin*θ* is the following:(5)$$\sin\theta =\theta -\frac{\theta^{3}}{3!}+\frac{\theta^{5}}{5!}+\ldots$$

Equations (2) and (4) are transformed into Equation (5), which neglects high-order terms due to *θ* being small enough:(6)$$l_{e}=\sqrt{1-\frac{1}{6}{\left(\frac{F}{2EI}l^{2}\right)}^{2}}l$$

The deflection at the free end can be modified as [38]:(7)$$\delta =\frac{l_{e}^{3}}{3EI}=\frac{1}{3EI}{\left(\sqrt{1-\frac{1}{6}{\left(\frac{F}{2EI}l^{2}\right)}^{2}}\right)}^{3}l^{3}$$

From Figure 2, the length of the long beams can be expressed as the following:(8)$$l_{0}=l\;l_{j}=2l+\left(2j-1\right)t\;\;\;\;\;\;\;\;\;\;\;\;\;\;\;\;\;\;\;\;\;\;\;\;\;j=1\sim n-1\;l_{n}=l+\left(n-1\right)t$$
where *t = p ×* tan*φ.* Therefore, the equation of the nonlinear relationship between the force and displacement for the ASSM is the following:(9)$$\delta_{T}=\overset{n}{\underset{j=0}{\sum }}{\left(-1\right)}^{j}\frac{1}{3EI}{\left(\sqrt{1-\frac{1}{6}{\left(\frac{F}{2EI}l_{j}^{2}\right)}^{2}}\right)}^{3}l_{j}^{3}$$

An ASSM can be made through silicon-based micromachining and measured force and displacement by a Microforce Testing System machine [34]. The fabrication processes and measurement are expressed in the Appendix A.

The microsprings were successfully fabricated, as presented in Figure 4. The scanning electron microscope (SEM) photograph revealed that the springs had good sidewall performance and a rectangular cross-section.

## 3. Results

A comparison of the experimental and theoretical results is presented in Figure 5a,b in which *N* = 8, *p* = 80 μm, *l* = 200 μm, *w* = 40 μm, and *h* = 100 μm. The coefficients of the determination (r-squared, *R^2^*) for Figure 5a,b are 0.982 and 0.945 respectively. The linear model fits well the experimental data. In these figures, the linear theoretical models are cited from [34]. The nonlinear models are from Equation (9). As illustrated in Figure 5, the theoretical results agreed well with the experimental results. The nonlinear relationship between the force and displacement for the ASSMs occurs when the applied force is large. From the results, the nonlinear relationship between the displacement and force for the ASSM was similar to that of a hardening spring, as expressed in Equation (1). Therefore, the ASSM had a similar nonlinear force–displacement relationship to that of hardening springs found in springs made from ductile material.

## 4. Discussion

To investigate the nonlinear relationship between the displacement and force for ASSMs, when the nonlinear displacement occurs, it must be defined. This can be defined as the percentage of nonlinearity that occurs when the displacement of the springs between the linear and nonlinear theoretical results differs [39]. Therefore, in this work, nonlinearity occurred when the differences between the linear and nonlinear displacement results were 10%, as shown in Figure 6, in which the difference in displacement between the linear and nonlinear results was 10%:(10)$$\frac{\delta_{nonlinear}-\delta_{linear}}{\delta_{nonlinear}}\times 100\%=10\%$$

Both the displacements of *δ_nonlinear_* and *δ_linear_* are functions of applied force *F*, and *δ_nonlinear_* = *δ_nonlinear_(F)* is calculated from Equation (9). From Equation (7) [34], *δ_linear_* = *δ_linear_(F)* is obtained. The corresponding applied force *F* was defined as the critical force in this work. Therefore, by using the critical force, the geometric effect on nonlinear displacement in the ASSM could be discussed.

Due to *δ_nonlinear_* in Equation (10) being from Equations (8) and (9), the effect of the taper angle, a parameter shown in Equation (8), on the nonlinear displacement of the ASSM is presented in Figure 7, which shows the relationship between the taper angle and critical force in which *N* = 8, *p* = 80 μm, *l* = 200 μm, *w* = 40 μm, and *h* = 100 μm. As displayed in Figure 7, when the taper angle increases, the critical force decreases nonlinearly. Therefore, the effect of the taper angle on the nonlinear deformation of ASSMs is significant. In this figure, critical force decays quickly when the taper angle is small. However, as the taper angle grows, the decreasing rate of critical force decays slowly. It is noticed that, from Equation (8), the length of the long beams is *l_j_ = 2l + (2j − 1) × p ×* tan*φ*. The larger taper angle would induce larger *l_j_*. Therefore, the nonlinear displacement would happen more easily for the larger taper angle due to the larger length of the long beams. Figure 8 illustrates the effect of the number of turns on the nonlinear displacement of ASSMs. This figure shows the results of the critical force for the different numbers of turns and the taper angle in which *p* = 80 μm, *l* = 200 μm, *w* = 40 μm, and *h* = 100 μm. As the number of turns increases, critical force decreases. The decreasing rate of critical force is more obvious for the larger taper angles. If the ASSM has a taper angle of 0° (the 0°ASSM is a traditional serpentine microspring), the critical forces are the same for all numbers of turns. Thus, the number of turns has no effect on the nonlinear displacement of traditional serpentine microsprings. Therefore, the effect of the number of turns on nonlinear displacement for the ASSM is nonsignificant when the taper angle is small. The effect of *l* on the nonlinear displacement of the ASSM is presented in Figure 9, which shows the relationship between *l* and the critical force where *N* = 8, *p* = 80 μm, *φ* = 20°, *w* = 40 μm, and *h* = 100 μm. In Figure 9, *l* is the shortest beam of the horizontal beam, *l*_0_. The critical force decreases nonlinearly as *l* increases. For the longer *l*, the nonlinear displacement of the ASSM happens easily. When compared to what is presented in Figure 7, the nonlinear decay rate of the critical force due to *l* is smaller than the decay rate from the taper angle. Therefore, the effect of the taper angle on the nonlinear displacement of the ASSM is more significant than *l*.

## 5. Conclusions

The ASSMs made by brittle materials used in MEMS were found to have a nonlinear displacement–force relationship similar to traditional ductile material springs. The nonlinear displacement–force relationship is due to the geometry and dimensions of ASSMs. The geometric effect of the nonlinear displacement–force relationship of ASSM was investigated. A theoretical solution was derived. ASSMs were successfully fabricated and measured. The theoretical results agreed well with the experimental results. The ASSM has a nonlinear displacement–force relationship similar to that of traditional springs made from ductile material. The results revealed that the taper angle had a significant effect on the nonlinear displacement–force relationship of ASSMs. However, the number of turns had no obvious effect on the nonlinear displacement–force relationship of ASSMs when the taper angle was small. Longer beam lengths facilitated nonlinear displacement.

## Figures and Tables

**Figure 1 materials-13-02864-f001:**
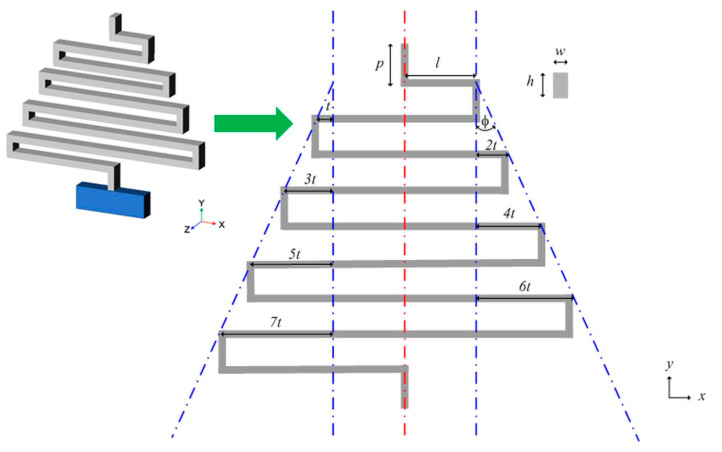
Awl-shaped serpentine microspring (ASSM) model.

**Figure 2 materials-13-02864-f002:**
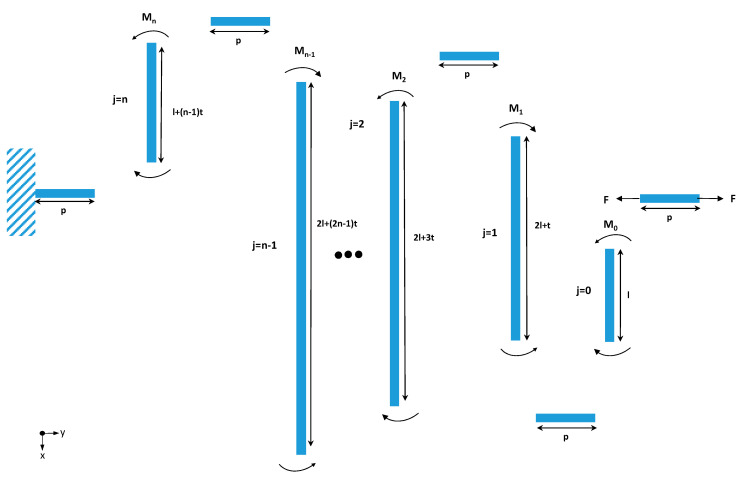
Free body diagram of the ASSM.

**Figure 3 materials-13-02864-f003:**
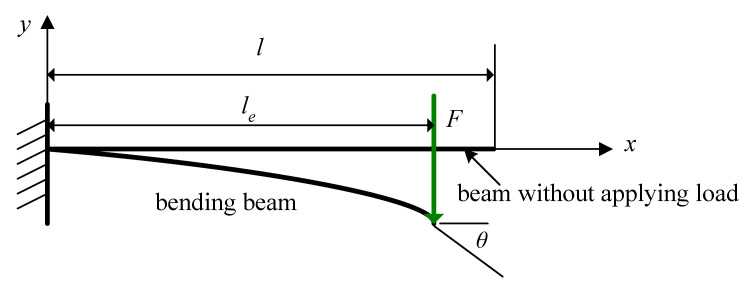
Bending a cantilever beam.

**Figure 4 materials-13-02864-f004:**
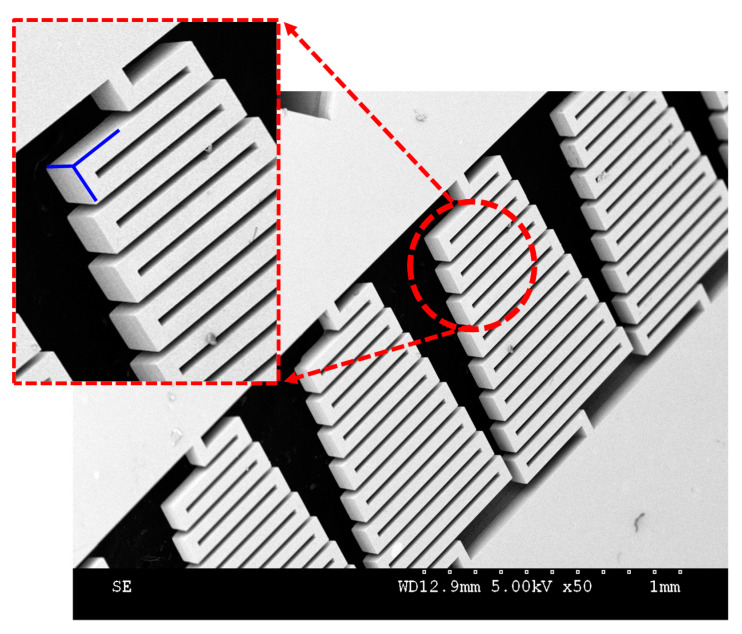
SEM photograph of awl-shaped serpentine microsprings.

**Figure 5 materials-13-02864-f005:**
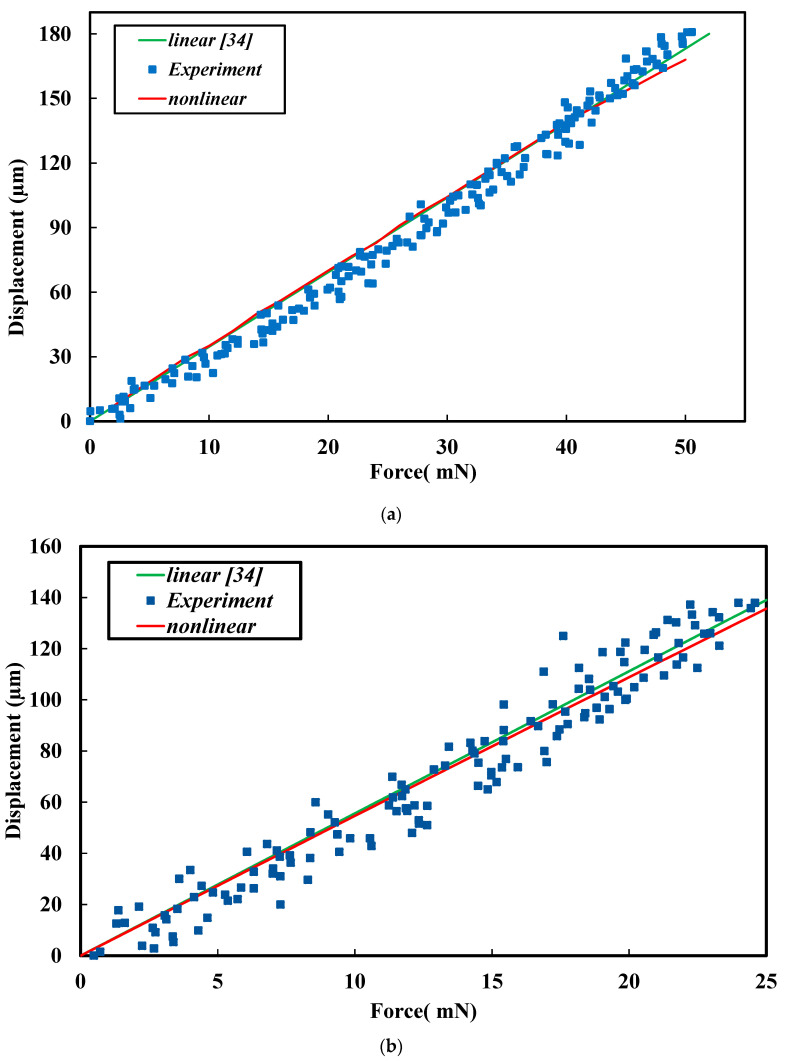
Comparison of linear and nonlinear models with experimental results in which *N* = 8, *p* = 80 μm, *l* = 200 μm, *w* = 40 μm, and *h* = 100 μm. (**a**) *φ* = 20°, (**b**) *φ* = 30°.

**Figure 6 materials-13-02864-f006:**
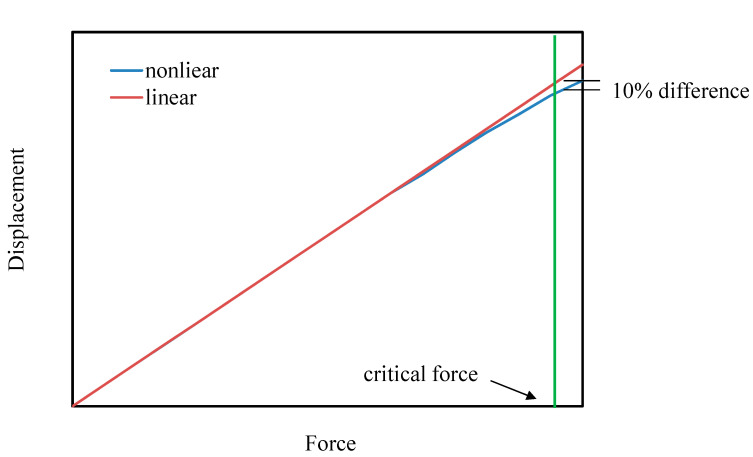
Definition of critical force for the ASSM.

**Figure 7 materials-13-02864-f007:**
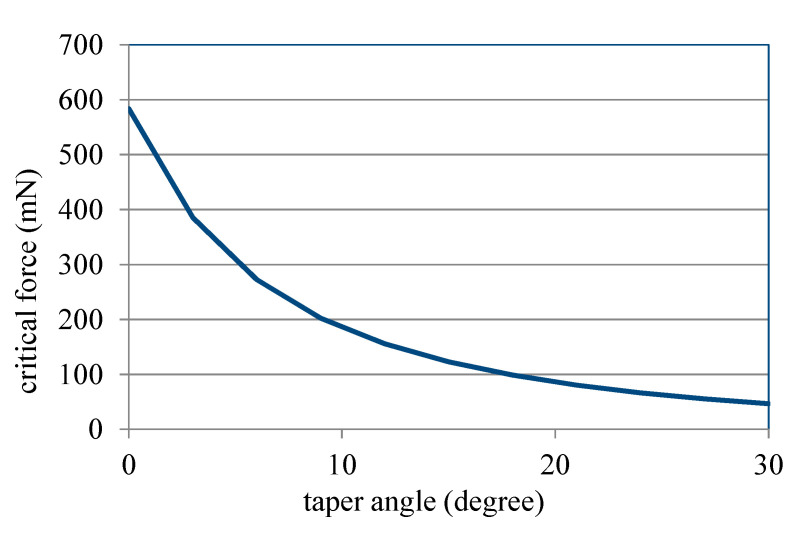
Relationship between the taper angle of ASSM and critical force calculated from Equation (10).

**Figure 8 materials-13-02864-f008:**
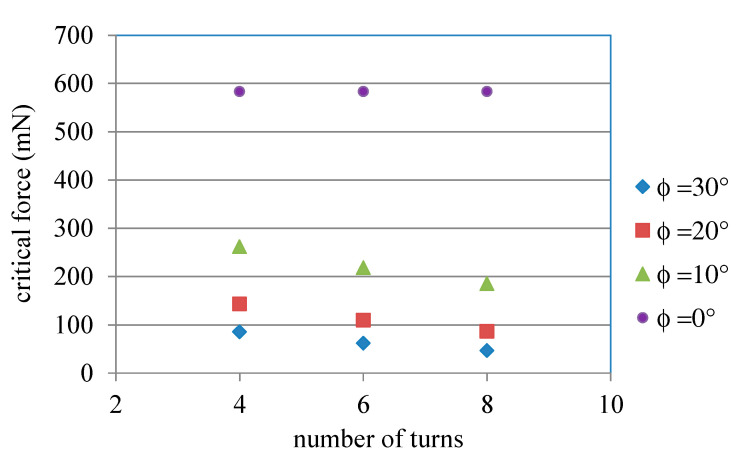
Relationship between the number of turns and critical force for ASSM calculated from Equation (10).

**Figure 9 materials-13-02864-f009:**
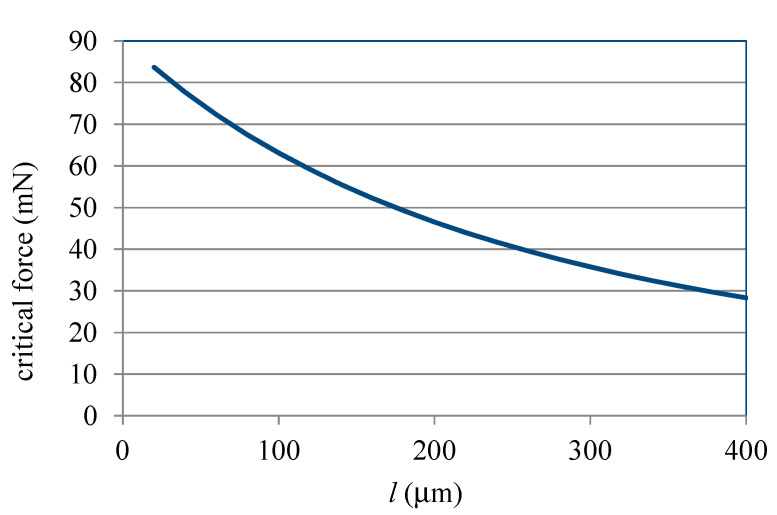
Relationship between *l* and critical force for ASSM calculated from Equation (10).

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
