# Peer review of "Geometric Effect on the Nonlinear Force-Displacement Relationship of Awl-Shaped Serpentine Microsprings for In-Plane Deformation"

_materials, 2020, doi:10.3390/ma13122864_

Round 1

Reviewer 1 Report

MDPI Material- 835608

Geometric effect on the nonlinear force–displacement relationship of awl-shaped serpentine microsprings 4 for in-plane deformation

The manuscript discusses the effect of different geometrical parameters including the taper angle, beam length, and the number of turns on the non-linear deformation of an owl-shaped spring. The force at which the non-linear displacement happens was marked where the difference between the linear and non-linear displacement graphs gets more than 10%. The manuscript needs some modifications before being suitable for publishing.

  1. Page 2, Line 67-68; Consider re-phrasing this sentence or adding commas:

“Traditionally, springs made from ductile materials such as metal present nonlinear behavior such as hardening the spring.”

  1. Page 8, Line 199; How does Equation (10) present the effect of taper angle on the nonlinear displacement of ASSM?

“From Equation (10), the effect of taper angle on the nonlinear displacement of ASSM is presented in Figure 9, which shows the relationship between taper angle and critical force…”

  1. Page 8, line 201; Could authors comment on the reason why the critical force decreases nonlinearly with increasing the tapper angle?

“As displayed in Figure 9, when the taper angle increases, the critical force decreases nonlinearly. Therefore, the effect of taper angle on the nonlinear deformation of ASSMs is significant. In this figure, critical force decays quickly when the taper angle is small. However, as the taper angle grows, the decreasing rate of critical force decays slowly. ”

  1. In the caption of graphs of Figures 9, 10 and 11, please indicate if they are calculated based on theory or experiment? 
  2. In Equation (2), how is the moment of inertia calculated for the owl-shaped serpentine spring?
  3. Page 3, line 93; In the manuscript, it is mentioned that the in-plane displacement of the owl-shaped spring is being investigated

 “only the y-direction deflections of long beams from the bending moment were considered.”

However, the bending beam model of Figure 3, shows an out-of-plane displacement of a cantilever. Considering the fixed end of the cantilever, the displacement of the cantilever is not of the same type as the displacement of the owl-shaped beam. How do the equations used for the bending cantilever could be justified for the in-plane motion of the owl-shaped beam?

Reviewer 2 Report

Nonlinear deflection analysis of micro awl-shaped serpentine springs for in-plane deformation

Geometric effect on the nonlinear force–displacement relationship of awl-shaped serpentine microsprings for in-plane deformation

  • Summary

This article describes awl-shaped serpentine microsprings (ASSM) from microfabrication to stress – strain experiment and comparison to a non-linear analytical model obtained using Taylor series. The work includes a description of the microfabrication of the ASSM then a description of mechanical response to in-plane tensile forces.

  • Overall opinion

This article still has many issues, the biggest one being its lack of originality and now the lack of improvement compared to previous submission. The language still needs to be drastically improved. As the microfabrication and experimental seem already published (doi: 10.1117/1.JMM.15.3.035003 and 10.1016/j.proeng.2016.11.300), the only original contribution seems to be the analytical model which complement the finite element approach of the former study and the comparison with the former results. Figures from former articles should not be reproduced.

  • Major Comments

Introduction:

The language has been slightly improved, the biggest change being the l.62-72 paragraph. There is a missing connection with the previous paragraphs.

Models:

This analytical model, which is one of the few original contribution of this work, has many issues among which:

l.80 according to Fig 1, it seems t/2p = tan(phi)

The way the angle theta  is defined in Fig. 3, the projection should be l = ltheta cos(theta)  (for theta = 0 ,l = ltheta).

Equation (2) and (3) are the same. Equation (3) could be simplified (squareroot in particular). It does not help to understand the connection between (2) and (4) injected in (5) to obtain (6).

Fabrication and measurement

Why not calling this section Material and methods?

Is this section different from former articles? If no, it should simply be referred to former articles.

Results and discussion

The experiments still miss reproducibility (displaying error bars and number of samples tested). Figures’ caption should be more specific i.e. giving more experimental details (such as for which parameters it has been made).

Fig 7 should be a horizontal facet plot with the same vertical scale. It might also display the fitting values and R2.

The critical force is now defined. However, Fig 8 caption is misleading. To my understanding this figure is only here to provide a definition of critical force. If so, the authors should just remove it.

The figure 9, 10 and 11 share the same vertical scale, however the critical force magnitudes are drastically different. They should share the same experimental parameters as much as possible.

There is not much discussion on the results. Particularly, why is it important to be able to tune the hardness of springs? Is there applications and alternative approaches?

The manuscript should stick to the guidelines of the journal:

  • Research manuscript sections: Introduction, Results, Discussion, Materials and Methods, Conclusions (optional)
  • It is important to make datasets available.

Reviewer 3 Report

The present paper discusses the geometric effect on the non-linear force-displacement relationship of ASSM spring for in-plane motion. The paper seems to be interesting and can be accepted after addressing the below mentioned comments.

  1. The abstract need to be revised by adding more conclusive advantages derived from this study.
  2. The article motivation is rather weak and unclear.
  3. Compare this research’s findings with state of the art and explain the benefits of this approach.
  4. Conclusion should be revised based on some scientific explanation.

Reviewer 5 Report

In the paper entitled “Geometric effect on the nonlinear force–displacement relationship of awl-shaped serpentine microsprings for in-plane deformation”, Authors present the results of relationship of awl-shaped serpentine microsprings for in-plane motion was investigated.  Presentation results is clear and contains interesting information. Article is very interesing.

The paper could be published after the following comments are addressed. Detailed comments are given below:

1) Please describe the production process exactly. There are no parameters as well as no precise description of the methodology.

2) Figure 4 is not well understood (please correct it).

3) The introduction does not include a specific application of the material.

4) Please extend your conclusions - point out the most important observations.

After considering the suggestions, the article is printable.

Round 2

Reviewer 2 Report

Nonlinear deflection analysis of micro awl-shaped serpentine springs for in-plane deformation – version 2

Line numbering of the v2 manuscript has missing lines (example: l.187-258). It is a rather important problem as I don't see most of the updates mentioned in the replies to the different reviews (example: "Our answer: (...) which shows the relationship between taper angle and critical force in which…….” It can be found in lines 193 196.")

The language has been improved, the structure and the contribution of this article is clearer. The scientific contribution is still low but the authors are not able to perform new experimental work.

Introduction:

There is still a sentence missing to connect l.57 to l. 58.

Models:

The model is now easier to access (and to relate to the following figures).

However, it is rather paradoxical to use a model for the large displacement (ref 37 Chen et al. ) and then suppose small angles hypothesis for Taylor development. Could the authors extend the discussion here?

Results and discussion

Though Fig 7 does not any estimation method (such as least squares) to predict the parameters of the model, the authors compare a linear model to experimental data. The R2 could be obtained and provide a quantity to assess how the linear model fits well the experimental data.

Also, the authors should rather name the curves “linear and non linear models” rather than “results” which remains vague and could let think the data is experimental.

Author Response

Reviewer’s comments:

Line numbering of the v2 manuscript has missing lines (example: l.187-258). It is a rather important problem as I don't see most of the updates mentioned in the replies to the different reviews (example: "Our answer: (...) which shows the relationship between taper angle and critical force in which…….” It can be found in lines 193 196.")

The language has been improved, the structure and the contribution of this article is clearer. The scientific contribution is still low but the authors are not able to perform new experimental work.

Our answer: Thanks for reviewer’s comment. It is due to revisions being highlighted the "Track Changes" function in Microsoft Word. And the highlights by “Track Changes” are not removed in pdf file. We have re-summited a pdf file without “Track Changes”.

Reviewer’s comments:

Introduction:

There is still a sentence missing to connect l.57 to l. 58.

Our answer: Thanks for reviewer’s comment. We add a sentence “For in-plane motion, the parameter spring constant to layout area ratio (k/A) is used as the performance index for ASSMs.” to express clear. Please see lines 55-56.

Reviewer’s comments:

Models:

The model is now easier to access (and to relate to the following figures).

However, it is rather paradoxical to use a model for the large displacement (ref 37 Chen et al. ) and then suppose small angles hypothesis for Taylor development. Could the authors extend the discussion here?

Our answer: Thanks for reviewer’s comment. In reference [37], large deflection of cantilever beam is discussed. Their model can be used for ductile and brittle materials. For ductile materials, the rotating angles would reach to 90 degrees when the deflection is very large. However, the brittle materials cannot be bent to induce very large deflection. Based on their model, the angle is small and able to use Taylor development.

Reviewer’s comments:

Results and discussion

Though Fig 7 does not any estimation method (such as least squares) to predict the parameters of the model, the authors compare a linear model to experimental data. The R2 could be obtained and provide a quantity to assess how the linear model fits well the experimental data.

Also, the authors should rather name the curves “linear and non linear models” rather than “results” which remains vague and could let think the data is experimental.

Our answer: Thanks for reviewer’s comments. We add the sentences “The coefficients of determination (r-squared, R2) for Figure 5 (a) and (b) are 0.982 and 0.945 respectively. The linear model fits well the experimental data.” As shown in lines 164-166. And we have renamed the curves “linear and nonlinear models” as shown in line 165.

This manuscript is a resubmission of an earlier submission. The following is a list of the peer review reports and author responses from that submission.

Round 1

Reviewer 1 Report

The manuscript is discussing the effect of different geometrical parameters including the taper angle, beam length, and the number of turns on the non-linear motion of an owl-shaped spring.

The writing of the manuscript should be improved to be more understandable. There are grammatical errors in the manuscript. The major point of the paper is discussing the effect of different geometrical parameters on the nonlinearity of the motion, which is mostly investigated by theory. The authors are asked to clarify what was the point of fabricating the spring? Since the experimental result shown in the graph of Figure 7, is the relationship between the force and displacement of the spring which is mostly in the linear regime. The other results are derived from theory. In Figure 7, how is the red graph (non-linear result) calculated? Is that calculated based on Equation 7? A discussion of why a larger force makes the displacement nonlinear should be added to the manuscript. How is the moment of inertia calculated for the owl-shaped serpentine spring? The results presented in the graphs of Figures 8, 9 and 10 should be justified by theory. The results and discussion section is only explaining what is shown in the graphs and is not giving any reason why is that happening.

Reviewer 2 Report

Nonlinear deflection analysis of micro awl-shaped serpentine springs for in-plane deformation

Summary

This article describes awl-shaped serpentine microsprings (ASSM) from microfabrication to stress – strain experiment and comparison to a non-linear analytical model obtained using Taylor series. The work includes a description of the microfabrication of the ASSM then a description of mechanical response to in-plane tensile forces.

Overall opinion

This article has many issues, the biggest one being its lack of originality. As the microfabrication and experimental seem already published (doi: 10.1117/1.JMM.15.3.035003 and 10.1016/j.proeng.2016.11.300), the only original contribution seem to be the analytical model which complement the finite element approach of the former study. Figures from former articles cannot be reproduced.

Major Comments

This analytical model has many issues among which:

1: For theta = 0, whereas we would expect the projection to be l The connection between equations (1) and (2) into (3) is non trivial Equation (7) does not relate to tensile force, and the reader cannot see the connection with the rest of the figures

The experiments should display reproducibility (displaying error bars and number of samples tested). Figures’ legend should be more specific.

The English level could be drastically improved.

The term “critical force” seems central but is not clearly defined. Is it a critical load or ultimate tensile strength? The following sentence (l. 158) does not help to define it.

“And the corresponding applying force is defined the critical force”

The manuscript should also follow the guidelines of the journal:

Research manuscript sections: Introduction, Results, Discussion, Materials and Methods, Conclusions (optional) It is important to make datasets available.

Reviewer 3 Report

In the present paper, the effect of nonlinear deformation of awl-shaped serpentine micro-spring is investigated and a theoretical solution is derived. The awl-shaped serpentine micro-springs has been successfully realized and the theoretical results agree well with experimental results. 

The paper is clear and well written, however, there are some issues that could be removed before the publication.

In particular, the conclusion paragraph has to be improved and extended.   

Furthermore, I, personally, suggest adding a list of acronyms and abbreviations. 

Moreover, I suggest to revise the English Language, there are many typos and misspellings in the paper.